# Strain and Shear-Wave Elastography and Their Relationship to Histopathological Features of Canine Mammary Nodular Lesions

**DOI:** 10.3390/vetsci9090506

**Published:** 2022-09-15

**Authors:** Marcella Massimini, Alessia Gloria, Mariarita Romanucci, Leonardo Della Salda, Lucia Di Francesco, Alberto Contri

**Affiliations:** 1Faculty of Veterinary Medicine, University of Teramo, Loc. Piano d’Accio, 64100 Teramo, Italy; 2Faculty of Biosciences and Technologies for Agriculture Food and Environment, University of Teramo, Via Balzarini 1, 64100 Teramo, Italy

**Keywords:** bitch, mammary tumour, strain elastography, shear-wave elastography, histopathology, fibrosis

## Abstract

**Simple Summary:**

Canine mammary tumours are the most represented neoplastic disease in canine medicine with relevant implications for the health of dogs. Diagnostic imaging offers tools able to better define lesion characteristics to improve treatment strategies. In recent years, sonoelastography, derived from classical ultrasonography, was studied for its ability to quantify in a semi-quantitative or quantitative manner the stiffness of tissue. In this study, two different methods, one semi-quantitative, namely strain elastography, and one quantitative, namely shear-wave elastography, were compared for their ability to quantify the properties of naturally onset canine mammary tumours. Shear-wave elastography was found to be more replicable, and both techniques were correlated with the amount of connective tissue of the lesion, suggesting that this attribute is largely responsible for the stiffness of the mammary lesion. Both elastographic techniques, however, were not able to distinguish between benign and malignant mammary tumours, which had a wide variable content in the connective tissue. The findings suggest that sonoelastography is useful for characterizing connective tissue content in the canine mammary tumour but should be used in conjunction with other techniques to define the malignancy of the lesion.

**Abstract:**

Mammary gland tumours have a significant impact on the health of dogs, requiring diagnostic tools to support clinicians to develop appropriate therapeutic strategies. Sonoelastography is an emerging technology that is able to define the stiffness of the tissue and has promising applications in the evaluation of mammary gland lesions. In the present study, strain elastography (STE) and shear-wave (SWE) elastography were compared in 38 mammary nodular lesions for their ability to define the histopathological features of canine mammary lesions. Among the techniques, SWE showed better repeatability (intraclass correlation coefficient: 0.876), whereas STE was found to be only acceptable (intraclass correlation coefficient: 0.456). Mammary nodular lesions showed a wide range of tissue stiffening with a similar mean value for STE and SWE in benign (4 ± 0.3 and 115.4 ± 12.6 kPa, respectively) and malignant lesions (3.8 ± 0.1 and 115.5 ± 4.5 kPa, respectively). A significant correlation was found between lesion fibrosis and STE (STE-I: r = 0.513, *p* < 0.001; STE-R: r = 0.591, *p* < 0.001) or SWE-S (r = 0.769; *p* < 0.001). In conclusion, SWE was reliable and correlated with fibrosis and was similar for both benign and malignant lesions, suggesting that other collateral diagnostic techniques should be considered in conjunction with SWE to characterize mammary nodular lesions in dogs.

## 1. Introduction

Mammary gland tumours represent the most relevant neoplastic disease in intact female dogs, with a reported incidence rate in Europe of between 0.2% [1,2] and 0.3% [3], which is higher than that seen in the human population [4]. Thus, specific attention has been devoted to early diagnosis and prompt treatment [5], aimed at preserving canine health. Due to similar clinicopathological and molecular features, canine mammary tumours (CMTs) are considered a valuable model of study for human breast cancer [6], although relevant differences exist in the hormonal cycle and incidence rate of the various histological subtypes between canines and humans. In this respect, myoepithelial cell proliferation is much more common in CMTs, occurring in >20% of canine tumours compared to <0.1% of human tumours [7,8]. Unfortunately, detailed information on subtype classifications and grading of mammary tumours is only obtained by histology, with details usually not available before developing a treatment plan.

Ultrasonography appeared to be the tool of choice for the evaluation of CMTs due to its non-invasive and well-tolerated features. The B-mode ultrasound technique was proposed for CMT evaluation since some tumour features, such as irregular shape, poorly defined margins, heterogenic echotexture, and vertical orientation of the lesion, are conventionally indicative of malignancy [9,10]. The lack of specific B-mode ultrasound features for differentiating benign and malignant CMTs has been reported [11], making this technique of limited prognostic value in this species. In recent years, sonoelastography, a relatively new ultrasonographic technique, has been introduced as a diagnostic tool for soft tissue in human and domestic animal medicine, due to its ability to define the stiffening of the lesion based on its elastic properties [12,13]. Currently, two types of sonoelastography are available namely, strain elastography (STE) and shear-wave (SWE) elastography. STE is based on the measurement of tissue echoes before and after compression, mechanically and manually induced by the operator [14,15,16]. The mechanical properties of tissues are qualitatively displayed in real-time in a colour-graded manner, interpreting the features of the lesion both qualitatively, using a five-grade classification [14], or semi-quantitatively, by comparing the stiffening of the lesion with the normal surrounding tissue [17]. On the other hand, SWE is based on lateral acoustic radiation spanning the examined tissue, which propagates faster in hard tissues than in soft tissues [18]. The ultrasound module records the velocity of the propagation, allowing the quantitative measurement of tissue elasticity in real time, generally reported as kilopascal (kPa) or m/s [19], and in an operator-independent manner [20].

In human medicine, sonoelastography was proposed to define the cutoff values to discriminate between benign and malignant neoplasm, but univocal thresholds were not defined [16,21,22,23]. Despite the lack of a broad consensus on the cutoff value for malignant and benign breast tumour discrimination, there is general agreement that harder lesions are related to malignancy [12,16,24,25].

Recently, different studies have evaluated the elastic properties of CMTs from a quantitative [26,27] or qualitative point of view [28]. However, to the best of our knowledge, no study exists comparing sonoelastographic findings obtained with both STE and SWE from the same lesion of a canine mammary gland. Thus, in the present study, STE and SWE were compared for their ability to detect different histological properties of spontaneous CMTs.

## 2. Materials and Methods

### 2.1. Patient Selection

This study was conducted between January and December 2020 on spontaneously occurring mammary gland nodular lesions from female dogs referred to the Veterinary Teaching Hospital of the University of Teramo, Italy.

All the ultrasound and surgery procedures were performed within the standard diagnostic and therapeutic procedures performed for this disease, and no unnecessary procedures were performed on the patients. Informed consent was obtained from the owners before their animals were included in the present study.

After a clinical examination, the mammary glands were carefully palpated and the side (left or right) and localization (cranial thoracic, caudal thoracic, cranial abdominal, caudal abdominal, inguinal) of each lesion were recorded. Then, the animals were subjected to ultrasound, as well as complete haematological and clinical biochemical evaluations. Furthermore, B-mode ultrasounds of the abdominal organs and radiological evaluations of the thorax in three projections were performed to exclude detectable metastasis. Only animals with haematological normal values and without clinical evidence of pulmonary and/or abdominal metastases were included in the study. A total of 22 canine patients met the criteria of inclusion and were considered for the ultrasound and histological evaluations of mammary gland nodular lesions. 

### 2.2. Ultrasound Sonoelastography

All ultrasonographic examinations were performed by the same operator with 3years of experience in the use of STE and quantitative SWE.

In this study, all mammary nodules were analyzed by both semi-quantitative STE and quantitative SWE using Logiq S8 (GE Healthcare, Milwaukee, WI, USA), with a 9L-D multi-frequency linear probe (GE Healthcare). All examinations were performed at a frequency of 8 MHz.

For STE, the nodular lesion was detected in an ultrasound B-mode scan and then the STE tool was activated. Strain elastographic images were generated by freehand cycles of compression/decompression, maintaining the lesion within the ultrasound scan. For each lesion, at least three clips were recorded. The software for STE was implemented with feedback by which the adequate compression and quality were visualized on a scale from red (low quality) to green (good quality). Only images with good-quality feedback were considered and the analyses were performed on at least three images recorded on three different clips. A region of interest (ROI) was created on the lesion, and an ROI of equivalent size and depth was positioned on apparently normal mammary tissue (Figure 1). The strain elasticity index (STE-I), as the value recorded by the software in the ROI of the lesion, and STE ratio (STE-R), as the ratio between the strain elasticity in the lesion and that in the normal mammary tissue, were calculated and recorded.

After STE, SWE was performed using the same probe. The colour-scale ranged from 0, displayed in red (soft), to 150+ kPa, displayed in blue (hard). During acquisition, the operator minimized pressure and movements with the probe on the lesion. Images were acquired by keeping the probe stationary for at least 10 seconds on the lesion for each measurement, and the orientation of the probe was chosen taking care that the maximum diameter of the lesion was included in the image. In SWE mode, the SWE image and B-mode image of the lesion were simultaneously visualized on the display in split-screen mode. For each mammary lesion, three clips were recorded and in each clip, three different frames were chosen to perform the measurements. Shear-wave rigidity was recorded on the ROI manually selected by the operator within the margins of the lesion (Figure 2). A second ROI, similar in size, was created on apparently normal parenchyma to measure the rigidity of normal mammary tissue. Areas with artefacts or without elastographic information (black-to-white areas in the image) were not considered. For each ROI, the shear-wave rigidity in kPa was measured and recorded (SWE-S).

Analyses of the images and recording of the STE and SWE parameters were performed by the same experienced operator who was not made aware of the results of the histological evaluation.

### 2.3. Surgery of Canine Mammary Nodular Lesions

After ultrasound evaluation, patients underwent a total or regional mastectomy, depending on the location of the mammary lesion. Localization in the caudal abdominal or inguinal mammary gland resulted in the caudal regional resection (from the cranial abdominal to the inguinal mammary gland); localization in the cranial or caudal thoracic mammary gland resulted in the cranial regional resection (from the cranial thoracic to the cranial abdominal mammary gland); localization in the cranial abdominal resulted in the resection of the complete mammary line. Surgery was performed under general anaesthesia with conventional procedures, and post-surgery antimicrobial therapy (amoxicillin and clavulanic acid, 20 mg/kg BID orally, Synulox, Zoetis Italia s.r.l., Rome, Italy) was performed for 7 days together with anti-inflammatory therapy (Rimadyl, 2 mg/kg SID orally, Zoetis, Italy) for 5 days. Follow-up was performed every 6 months after surgery by radiological thoracic examination and ultrasound abdominal evaluation.

### 2.4. Histopathological Evaluation of Canine Mammary Nodular Lesions 

After surgery, the mammary nodules and surrounding mammary tissue (at least 2 cm) were fixed in 10% neutral-buffered formalin and embedded in paraffin wax. Histological classifications [29] and grading [30] were established using haematoxylin and eosin (H/E) stained slides. Additional sections were also stained with Masson trichrome for evaluating the extent of collagen deposition and Movat pentachrome stain for detecting mucin-rich/collagen-poor ECM (extracellular matrix) [31] (Figure 3) and were subjected to immunohistochemistry (IHC) using a primary antibody directed against vimentin (1:100 dilution, V9, mouse monoclonal; Dako) for detecting myoepithelial and stromal cells. Briefly, for IHC, sections were treated with citrate buffer solution 0.01 M pH 6.0 in a microwave for 15 minutes for antigen retrieval and with 5% bovine serum albumin and 5% normal goat serum for 15 minutes each to block unspecific binding sites. Labelling was subsequently detected using an ImmPRESS HRP Universal (horse anti-mouse/rabbit IgG) PLUS polymer kit (Vector Laboratories, CA, USA) with 0.1% hydrogen peroxide in a 3,3′-diaminobenzidine solution (Millipore Sigma, St. Louis, MO, USA) as the chromogen. Sections were finally counterstained with Mayer haematoxylin (Merck, Darmstadt, Germany).

The percentage of fibrosis and vimentin-positive cells was calculated on low-magnification (25× and 50×, respectively) images of coronal sections of the entire nodular lesion using the colour detection function of the Image J IHC Tool, whereas the percentages of other histopathological features (necrosis, chondroid or bone tissue, cystic spaces/tubular secretion) were calculated on live digital imaging using the area measure tools of the Leica Application Suite X (LAS X Version 02) software (Leica Microsystems CMS GmbH—Wetzlar, Germany).

On the other hand, the percentage of mucin-rich/collagen-poor ECM on Movat pentachrome-stained slides was semi-quantitatively assessed as follows: − (absent), + (≤10%), ++ (>10%–≤50%), +++ (>50%).

### 2.5. Statistical Design

Data were reported as the mean and standard error of the mean (SEM). The normal distribution of the data was tested using the Shapiro–Wilk test. Quantitative attributes were considered normally distributed (*p* > 0.05) for all parameters except for necrosis, chondroid and bone tissues, and cystic spaces/tubular secretion. For normally distributed parameters, the Levene test for equality of variances was also performed. The prevalences of the side (left or right) and localization (cranial thoracic, caudal thoracic, cranial abdominal, caudal abdominal, inguinal) were checked by the chi-square test. 

The repeatability of STE and SWE in repeated measures performed on the same lesion in different clips and images was verified using the intraclass correlation coefficient (ICC). This test returns values between 0 and 1, and poor reliability was defined with an ICC below 0.5, moderate reliability between 0.5 and 0.75, good reliability between 0.75 and 0.9, and excellent reliability above 0.9 [32]. For normal mammary tissue, both STE-I and SWE-S were verified for the effects of age, breed, and location using the general linear model (GLM) based on ANOVA.

In addition to a comparison of benign vs malignant lesions, malignant tumours were also sub-grouped based on the histological subtype (i.e., complex, mixed, and ductal-associated) and grade of malignancy. Differences in the percentages of fibrosis, vimentin-positive cells, necrosis, chondroid and bone tissues, cystic spaces/tubular secretion, and mucin-rich/collagen-poor ECM among these groups were compared using the GLM. When appropriate, Scheffè post hoc analysis was also used.

Correlations between the average STE-I, STE-R, and SWE-S and the histological features were determined by calculating Pearson’s or Spearman’s correlation coefficients depending on the normality of the distributions. The difference between the levels of mucin-rich/collagen-poor ECM and SWE-S was evaluated with ANOVA. Statistical analysis was performed using SPSS 17.0 software (SPSS Inc., Chicago, IL, USA), with *p* < 0.05 considered statistically significant.

## 3. Results

A total of 38 nodular lesions of the mammary glands were detected in 22 canine patients. The included breeds were mixed-breed (*n* = 6), Chihuahua (*n* = 2), Lagotto Romagnolo (*n* = 2), German sheepdog (*n* = 2), Maltese (*n* = 2), Pinscher (*n* = 2), Border collie (*n* = 1), Boxer (*n* = 1), English setter (*n* = 1), Scottish sheepdog (*n* = 1), Shih-Tzu (*n* = 1), and Yorkshire (*n* = 1). The mean age of the animals at diagnosis was 9.06 ± 1.9 years, with a range between 5.7 and 13.3 years. Mammary nodular lesions were localized at the inguinal (I), caudal abdominal (cauA), cranial abdominal (crA), caudal thoracic (cauT), and cranial thoracic (crT) mammary glands in 14 (36.8%), 15 (39.5%), 4 (10.5%), 4 (10.5%), and 1 (2.6%) cases, respectively, with a significant prevalence in I and cauA when compared with the other localizations (*p* < 0.05). Among the lesions, 14 were found in the right mammary line (36.8%) and 24 in the left mammary line (63.2%), with a larger prevalence in the left mammary line (*p* < 0.05).

Radiographic and ultrasonographic follow-ups at 6 months revealed the onset of pulmonary metastasis, not detected at the first evaluation, in two patients (9.1%) with a grade II and grade III carcinoma, respectively. 

### 3.1. Sonoelastography and Correlation between Elastographic Parameters

Normal mammary gland parenchyma showed homogeneous stiffening values among the various animals studied, without differences between the breeds, ages, and locations of the lesion (*p* > 0.05). The STE-I of normal mammary tissue was 0.5, whereas the objective SWE-S was 18.8 ± 1.01 kPa.

The repeatability of the STE-I was moderate, with an ICC of 0.456. The reliability was good when the STE-R was considered, with a value of 0.649 for the ICC. The stiffening of the mammary nodular lesions measured by the SWE-S showed high repeatability, with an ICC of 0.876. The STE-I, STE-R, and SWE-S of mammary nodular lesions recorded in this study are summarized in Table 1.

No significant correlations were found between the STE-I and SWE-S (r = 0.457; *p* > 0.05), whereas a weak but significant correlation was found between the STE-R and SWE-S (r = 0.684; *p* < 0.05).

The elastographic parameters (STE-I, STE-R, and SWE-S) in the hyperplastic/benign neoplastic lesions were similar to the malignant lesions (Table 1).

### 3.2. Histological Evaluations

The histologically examined nodular lesions (*n* = 38) of the canine mammary glands were classified into hyperplastic (*n* = 3), benign (*n* = 2), and malignant (*n* = 33) lesions and are summarized in Table 2.

Fibrosis, vimentin-positive cells, and cystic spaces/tubular secretion were variably recorded in all cases, whereas necrosis was detected in 30/38 cases and chondroid and bone metaplasia were detected in 10 and 3 cases, respectively. ECM evaluation was not available for 5/38 nodules due to technical issues. Although not reaching statistical significance, a tendency toward higher fibrosis was observed in the hyperplastic/benign neoplastic lesions compared to the malignant lesions (50.4% ± 11.8% of hyperplastic/benign neoplastic lesions vs 33.0% ± 3.3% of malignant lesions; *p* = 0.87) (Figure 4).

As expected, a significantly (*p* = 0.035) higher percentage of vimentin-positive cells was also observed in complex and mixed malignant tumours (37% ± 3.9%) with respect to other malignant histotypes (26.9 ± 2.7%).

### 3.3. Correlations between Elastographic Parameters and Histological Features of Canine Mammary Nodular Lesions

The correlation analysis revealed a significant Pearson correlation coefficient between fibrosis and the STE-I (r = 0.513; *p* < 0.001), STE-R (r = 0.591; *p* < 0.001), and SWE-S (r = 0.769; *p* < 0.001). No significant correlations were found between the sonoelastographic parameters, estimated with both techniques, and other histological features of the nodules, except for a tendency observed in the correlation between the percentage of bone metaplasia and SWE-S (Spearman’s rho = 0.295 *p* = 0.072).

## 4. Discussion

In the present study, the cases included were predominantly represented by malignant neoplastic lesions (86.84%), showing a higher prevalence than that usually reported in the literature, ranging between 30% and 50% [33,34]. Notwithstanding these results, similar findings were shown in other recent studies concerning CMTs in which the prevalences reported were 82% [27] and 71% [11]. These differences could suggest a progressive increase in the prevalence of malignant lesions of the canine mammary gland, as hypothesized in a longitudinal study in which it increased to 70% [3]. Furthermore, the present findings showed a predominant detection of mammary lesions in the caudal mammary glands, which is consistent with the literature [35]. Moreover, it should be noted that the present research is not an epidemiological study, thus the prevalence of canine mammary gland lesions could be influenced by other reasons, possibly related to the secondary-care nature of the veterinary teaching hospital.

To the best of our knowledge, this is the first study in which the reliability of different sonoelastograpic techniques was estimated in the canine species. The data presented here showed that SWE appeared to be more reliable than STE to evaluate canine mammary nodular lesions. A recent study showed an excellent intra-operator repeatability using SWE during the examination of an elastic phantom [36], and the lesser agreement found in the present study (ICC = 0.876) could be related to the heterogeneous nature of the spontaneously occurring canine mammary lesions. On the other hand, the higher repeatability of SWE compared to both the STE-I and STE-R suggests that several repeated measurements and the mean calculation are required to reliably estimate the stiffness by STE.

In the present study, a comparison of the stiffness estimated by STE and SWE performed in the same canine mammary nodular lesion was reported for the first time. Previous studies in human medicine compared the diagnostic performances of STE and SWE for their ability to differentiate benign and malignant breast lesions, demonstrating similar diagnostic performances [37,38,39]. In the present study, a relevant agreement was recorded between these techniques, suggesting the effectiveness of both STE and SWE for the characterization of canine mammary nodular lesions. 

In the patients included in the present study, malignant mammary nodular lesions showed similar stiffness compared with hyperplastic/benign neoplastic lesions. This finding is apparently in contrast with previous studies in human medicine in which malignant breast tumours generally showed increased hardness [12,16,24,25,40]. Similarly, a series of studies on CMTs reported high values of stiffness in malignant lesions [27,41]. Notwithstanding these results, it is important to mention that our findings could be due to the limited number of hyperplastic/benign neoplastic lesions, which were mainly represented by histological lesions typically characterized by a high degree of fibrosis, such as lobular hyperplasia with fibrosis, and simple, sclerosing adenoma. In this respect, fibrosis was the main histological parameter correlating with the STE-I, STE-R, and SWE-S (*p* < 0.001) in our study. In addition, the presence of other intralesional features, such as central necrosis or the mucinous matrix, could also affect the stiffness of neoplastic tissue, making it more difficult to discriminate between benign and malignant lesions [42]. In our study, no significant correlations were found between the sonoelastographic parameters and various histopathological features of the mammary nodules, except for a tendency toward a higher stiffness in malignant histological subtypes characterized by osseous metaplasia. In this respect, it is important to note that CMTs are typically characterized by a great heterogeneity of cellular and extracellular components, which may include epithelial and myoepithelial cells, mesenchymal components, such as cartilaginous and/or osseous tissue, as well as a variably abundant myxoid extracellular matrix. It is also important to consider that these components may be variably represented in different mammary nodules with the same histological diagnosis, e.g., complex and mixed carcinomas may be characterized by a variably extensive presence of myoepithelial and/or mesenchymal components. Furthermore, among malignant CMTs, particular histotypes, such as carcinoma arising in complex adenoma/benign mixed tumour, may be characterized in some cases by only small areas of malignant tumour, which could not influence the elastic properties of the nodular lesion, which are likely related to the mechanical properties of the prevalent component. Thus, similar to previous studies [28,43], the accuracy of elastography appears to be limited in differentiating between benign and malignant canine mammary lesions, as well as discriminating between the different histological subtypes or grades of malignancy, although the limited number of canine patients with high grades of malignancy (*n* = 2) made the statistical comparison of the grades difficult. Further studies with a higher number of cases will allow for a better evaluation of the diagnostic and/or prognostic performance of elastography and to establish whether an optimal cutoff value exists to differentiate between benign vs. malignant canine mammary nodular lesions and/or to identify more aggressive histotypes of CMTs. 

As previously mentioned, among the different features of the mammary nodular lesions included in the present study, a significant correlation between stiffness and percentage of fibrosis was recorded, with better results provided by SWE. These findings were consistent with those described in fibrous lesions of other organs since sonoelastography was originally applied to fibrotic degeneration of soft tissue organs such as the liver [44] or kidney [45]. A significant and linear correlation was found between SWE and the Picrosirius red-stained collagen amount in liver fibrosis experimentally induced in a rat model [46]. Similarly, sonoelastography was significantly correlated with collagen fibre content in pancreatic fibrotic lesions [47]. Therefore, the findings reported in the present study suggested the effective use of sonoelastography to test the fibrotic content of canine mammary lesions rather than to discriminate between benign and malignant mammary lesions. Connective tissue deposition could play a relevant role in mammary pathophysiology in both humans and animals. In humans, mammographic density, which was found to be variable depending on the amount of collagen, was reported as a relevant risk factor for malignancy [48]. In a recent study, collagen density was also found to be negatively correlated with patient survival in dogs [49]. Other studies, however, showed that the nature of the collagen deposition rather than its amount could drive the protective or permissive role in tumorigenesis. Maller et al. reported that high-density non-fibrillar collagen I appeared to be tumour-suppressive, whereas linear (fibrillar) collagen was correlated with tumour invasiveness [50] and reduced survival [51]. It was hypothesized that the increased crosslink between collagen fibres may increase the stiffness of the extracellular matrix, promoting the invasion of an oncogene-initiated epithelium [52]. On the other hand, the presence of a clear tumour-stromal boundary, with parallel-arranged collagen fibres correlated with the increased overall survival of canine patients [49]. Furthermore, numerically consistent studies including follow-ups are required to clarify the possible application of sonoelastography to evaluate collagen tissue orientation in canine mammary gland nodular lesions.

## 5. Conclusions

In the present study, SWE appeared to be more reliable than STE in quantifying the stiffness of canine mammary nodular lesions and was also significantly correlated with the percentage of fibrosis within the nodule. Since a clear increase in fibrosis was not found in malignant lesions compared to hyperplastic/benign neoplastic lesions, both SWE and STE did not appear to be able to discriminate between the different natures of the canine mammary nodules, which are typically characterized by a great heterogeneity of cellular and extracellular components that may variably influence nodular stiffness.

## Figures and Tables

**Figure 1 vetsci-09-00506-f001:**
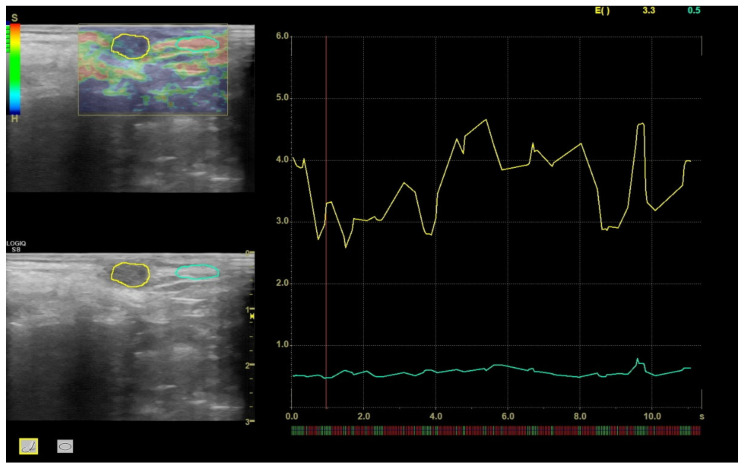
A representative image of the semi-quantitative measurements of STE-I and STE-R in a mammary nodular lesion in a bitch (sample No. 12d). The graduation scales reported in the left upper corner refer to the quality of the image acquisition (green bar on the left) and the different degrees of stiffness (red-to-blue bar on the right).

**Figure 2 vetsci-09-00506-f002:**
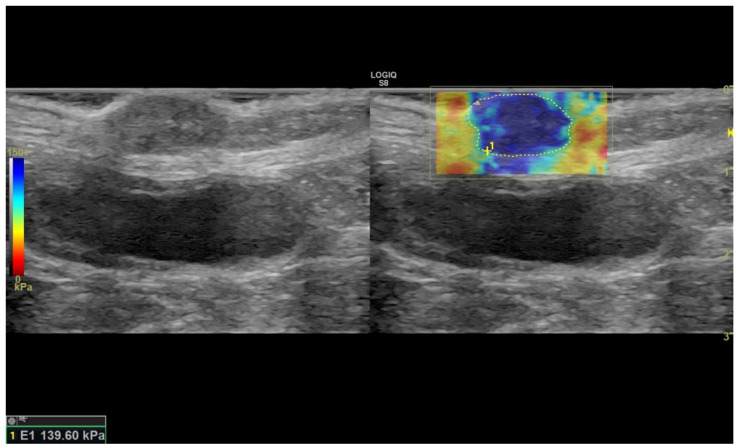
A representative image of the quantitative measurement of SWE-S in a mammary nodular lesion in a bitch was included in the study (sample No. 17). The graduation red-to-blue scales reported on the left of the image refer to the degree of stiffness. The region of interest (ROI) is delimitated by discontinuous yellow line (1).

**Figure 3 vetsci-09-00506-f003:**
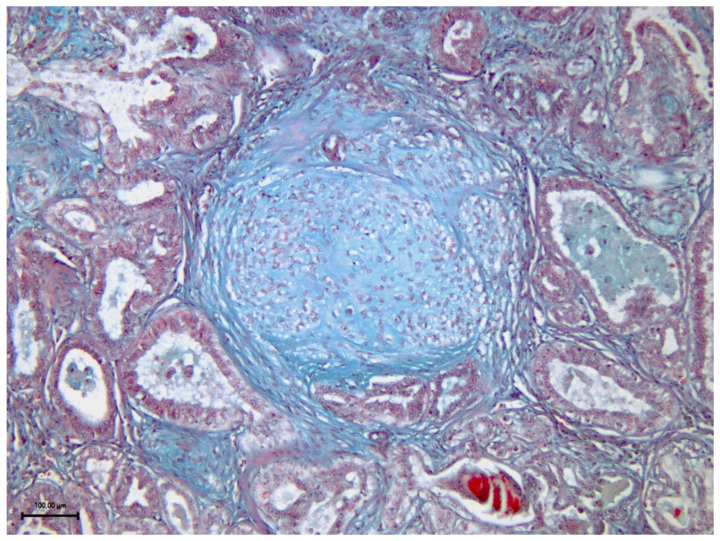
Mucin-rich/collagen-poor ECM in canine mammary lesions. Figure (100×) shows the light-blue ground substance of a myoepithelial nest in a mixed carcinoma (sample No.4) stained with Movat pentachrome.

**Figure 4 vetsci-09-00506-f004:**
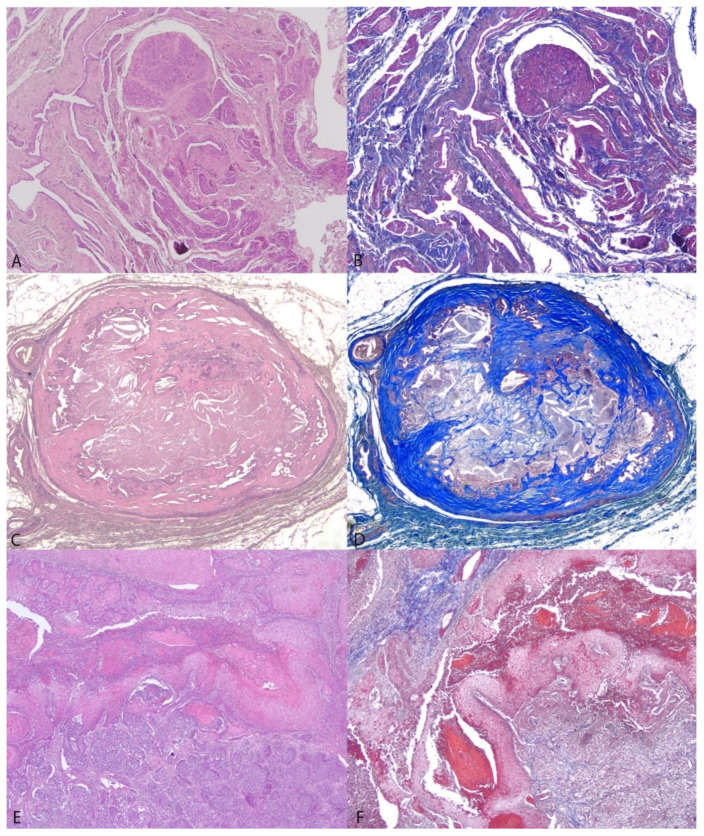
Fibrosis levels in benign and malignant canine mammary lesions. Figures (50×) show lobular hyperplasia with fibrosis (**A**,**B**: sample No. 8b), a sclerosing simple adenoma (**C**,**D**: sample No. 12b), and an adenosquamous carcinoma (**E**,**F**: sample No. 7b) stained with H/E in the left column (**A**,**C**,**E**) and Masson trichrome in the right column (**B**,**D**,**F**), where collagen fibres have been stained in blue.

**Table 1 vetsci-09-00506-t001:** Percentage of fibrosis and elastographic parameters in benign (hyperplastic and neoplastic benign lesions) and malignant (grade I, II, and III) canine mammary lesions (n. 38).

	Benign			Malignant			
	*Hyperplastic*	*Neoplastic*	Total	*Grade I*	*Grade II*	*Grade III*	Total
	*(n = 3)*	*(n = 2)*		*(n = 19)*	*(n = 11)*	*(n = 2)*	
**Age (y)**	8.9 ± 0.8	10.7 ± 2.7	9.6 ± 1	9.7 ± 0.5	9 ± 0.7	9.4 ± 2.6	9.4 ± 0.4
**Fibrosis (%)**	** *37.06 ± 15.59* **	** *70.50 ± 0.79* **	50.44 ± 11.84	** *32.35 ± 4.43* **	** *34.98 ± 6.13* **	** *34.52 ± 16.46* **	32.95 ± 3.32
**SWE-S (kPA)**	** *100.8 ± 15.8* **	** *137.4 ± 4.8* **	115.4 ± 12.6	** *114.9 ± 4.9* **	** *120.1 ± 6.7* **	** *88.7 ± 56* **	115.5 ± 4.5
**STE-I**	** *3.5 ± 0.3* **	** *4.7 ± 0.3* **	4 ± 0.3	** *4 ± 0.2* **	** *3.7 ± 0.2* **	** *3 ± 0.5* **	3.8 ± 0.1
**STE-R**	** *6.74 ± 1.82* **	** *7.64 ± 1.76* **	7.1 ± 1.16	** *5.69 ± 0.54* **	** *5.53 ± 0.6* **	** *4.46 ± 2.54* **	5.6 ± 0.38

Data are expressed as ***Mean ± SEM***.

**Table 2 vetsci-09-00506-t002:** Classification and histological features of the canine mammary gland nodular lesions included in the study.

N°	Histological Classification	Grade	Fibrosis (%)	Vimentin-Positive Cells (%)	Necrosis (%)	Chondroid (%)	Bone (%)	Cystic Spaces/Tubular Secretion (%)	ECM
1	Mixed carcinoma	II	12.2	36.7	0.5	0.15		4.9	++
2a	Carcinoma arising in benign mixed tumour	I	25.7	29.1	0.09	12.10		5.5	+++
2b	Tubular carcinoma	I	20.4	22.5	0.2			14.5	+
2c	Carcinoma arising in complex adenoma	I	10.8	28.3	20			1	+++
3	Comedocarcinoma	II	76.0	11.6	24			0.5	++
4	Mixed carcinoma	I	61.6	25.4	1.13	12.83	0.41	3.29	+++
5a	Intraductal papillary carcinoma	II	44.9	28.7	5.42			38.02	N.A.
5b	Complex carcinoma	I	6.2	50.0				0.85	+++
6	Intraductal papillary carcinoma	I	21.0	34.7				6.6	+++
7a	Complex carcinoma	II	3.8	51.0	0.57			1.02	++
7b	Adenosquamous carcinoma	III	18.1	23.3	3.91			0.26	N.A.
8a	Mixed carcinoma	I	11.4	27.2		11.43		9.75	N.A.
8b	Lobular hyperplasia with fibrosis	N.A.	67.4	10.1				0.23	N.A.
9	Intraductal papillary carcinoma	II	27.3	18.4	4.98			51.93	++
10	Complex carcinoma	I	27.8	34.3	0.33			20.33	+++
11	Lobular hyperplasia with atypia	N.A.	28.0	9.0				0.27	++
12a	Lobular hyperplasia with atypia	N.A.	15.8	12.4				18.98	++
12b	Simple adenoma, sclerosing	N.A.	71.3	20.0	9.33			3.04	++
12c	Intraductal papillary carcinoma	I	38.0	36.9	5.52			19.21	+++
12d	Carcinoma arising in benign mixed tumour	I	57.3	28.1	0.06	36.93	4.18	0.1	+
13a	Carcinoma arising in complex adenoma	I	35.2	27.1	0.45			26.08	+++
13b	Carcinoma arising in benign mixed tumour	I	22.2	43.5	0.57	9.93		1.6	+++
13c	Ductal carcinoma	I	33.5	63.1	0.43			2.46	++
14a	Intraductal papillary carcinoma	II	44.9	32.8	1.43			27.97	+++
14b	Intraductal papillary carcinoma	II	38.4	21.4	0.5			4.9	+++
15	Ductal carcinoma	I	30.8	12.4	0.04			3.25	+++
16	Ductal carcinoma	I	55.4	17.0				4.45	+++
17	Mixed carcinoma	I	22.8	24.2	0.12	0.44		37.42	
18a	Intraductal papillary carcinoma	I	21.0	25.0	0.8	3.25		25.96	++
18b	Carcinoma and- malignant myoepithelioma	III	51.0	34.6	13.52			1.02	++
18c	Intraductal papillary carcinoma	II	13.6	24.6	20			13.00	+
19	Complex carcinoma	II	35.0	16.1				1.29	+++
20	Carcinosarcoma	N.A.	18.9	27.2	2.05	26.25	13.18	3.41	+++
21	Multinodular: Lobular hyperplasia/simple adenoma/intraductal papillary adenoma–carcinoma/tubular–solid carcinoma	II	49.9	19.8	2.3			15.0	+
22a	Complex carcinoma	II	39.0	70.9	6.1			10.0	+
22b	Intraductal papillary carcinoma	I	32.4	41.3	0.07			13.8	+
22c	Multinodular: Lobular hyperplasia with secretory activity –with atypia/benign mixed tumour/simple adenoma/complex adenoma/intraductal papillary adenoma	N.A.	69.8	54.0	0.8	0.1		12.8	++
22d	Carcinoma arising in complex adenoma	I	81.3	50.0	0.22			2.22	N.A.

N° indicates the canine patient; letters indicate cases with multiple lesions. N.A.: not applicable.

## Data Availability

The data not presented in the manuscript are available for consultation after a reasonable request to the corresponding authors.

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
