# Peer review of "Strain and Shear-Wave Elastography and Their Relationship to Histopathological Features of Canine Mammary Nodular Lesions"

_vetsci, 2022, doi:10.3390/vetsci9090506_

Round 1
Reviewer 1 Report
This study entitled "Strain and shear-wave elastography and their relationship to histopathological features of canine mammary nodular lesions" has worked for the evaluation for canine mammary gland lesion by using the sonoelastography, STE and SWE. This work was well designed, and the methodology also was well presented. But the result section may need to be improved for a clear understanding.
1. Lines 78-79, the authors said there is some controversial finding in human medicine. The relevant description on the finding will be needed to be incorporated, and please insert the description of how the authors worked to overcome those limitations (i.e. controversial finding) when the sonoelastography was applied to canine cancer for this study in introduction or discussion part.
2. For this work, the samples from 12 different breeds were collected in the clinics. But no age information was provided although the authors mentioned "ages" as a variable in line 246. Please provide the information of any difference among breeds or ages with a statistical value (P-value and R) following the STE or SWE.
3. For table 1, total number of samples was missing. please add the numbers.
4. For table 2, histological features of 38 different nodule lesions were presented. Please add the information of each canine patient to see where the nodules were originated and sample number (No.) if possible.
5. For the figures (1 to 4), it seems that representative features were shown so can you add sample No. to identify the characteristics of the nodules clearly?
6. line.397, fix the spelling error.
Reviewer 2 Report
The study is well planned methodologically and is the first time that semiquantitative and quantitative elastography are objectively compared.
In contrast with other studies in canine and human medicine where stiffness increase with malignity here it has not been observed. As it is commented in limitations, the number of clinical cases can influence the results.
On the other hand, it is interesting to publish this article in order to increase the debate in scientific community to advance the knowledge.
A few typographic mistakes should be corrected, like the hyphen in the lines: 179, 180, 216, 267, 281, 374, 388.
Line 257: specify the acronym SH

Reviewer 3 Report
Comments on the manuscript VetSci-1883862 entitled “Strain and shear-wave elastography and their relationship to histopathological features of canine mammary nodular lesions” by Massimini et al.
The manuscript is an interesting investigation exploring image techniques on canine mammary tumor’s diagnosis. New articles on this theme are welcome.
The major limitation of this article is the small number of non-neoplastic (hyperplastic) and benign lesions. Another limitation is the absence of analysis of “non-neoplastic (“normal”) gland with no relation with tumours, as mammary gland adjacent to tumours suffers influences of tumour microenvironment. However, results are interesting… There are some changes that must be amended to increase comprehensibility.
Line 175- I think authors must clarify what they mean with “fibrosis”; fibrosis is not the presence of “collagen fibers”, as collagen is a normal and usual component of all tissue matrices. Fibrosis is “excess deposition of collagen”, not the presence of collagen. Please amend.
Line 178- Authors must amend the sentence …” for detecting non-epithelial (myoepithelial and stromal) cells.” as myoepithelial cells are epithelial cells with contractile properties. All histology books and the following reference mention “…The normal mammary gland is organized around a branched ductal network arranged in an epithelial bilayer, with an inner luminal and outer myoepithelial cell layer. Adriance et al (2005) …”
Replace the sentence to … “for detecting myoepithelial cells and non-epithelial (stromal) cells.”
-In all the manuscript there are multiple “hyphens “that must be removed (as in line 179 the word “so-lution”).
- The magnification of the histological images should also be amended, as authors just refer the objective magnification of the light microscopic, and should use the “ocular x objective”; when author mention “2.5x, 5x, 10x amplifications”, it should be replaced by “25x, 50x, 100x”, and so on.
Line 202- Amend E/E to H&E.
Line 224- “p value” should be written in lower case (p<0.05).
Line 241-242- Please mention that grade 2 and grade 3 lesions are carcinoma lesions, as grade is just applied to mammary carcinoma.
Line 261- Please remove “solid” from the sentence; sentence should be amended to mention “non-neoplastic or hyperplastic” and benign lesions.
Table 2- Amend “Complex carcinoma complesso”.
Round 2
Reviewer 1 Report
Thank you for the responses.